# Novel Ultrafast Lu$_2$O$_3$:Yb Ceramics for Future HEP Applications

Chen Hu [1], Liyuan Zhang [1], Ren-Yuan Zhu [1,*], Lakshmi Soundara Pandian [2], Yimin Wang [2] and Jarek Glodo [2]

1 High Energy Physics, California Institute of Technology, Pasadena, CA 91125, USA
2 Radiation Monitoring Devices, Inc., Watertown, MA 02472, USA
* Correspondence: zhu@caltech.edu

**Abstract:** Inorganic scintillators activated by charge transfer luminescence Yb$^{3+}$ are considered promising ultrafast material to break the _ps_ timing barrier for future high energy physics applications. Inorganic scintillators in ceramic form are potentially more cost-effective than crystals because of their lower fabrication temperature and no need for aftergrowth mechanical processing. This paper reports an investigation on Lu$_2$O$_3$:Yb and Lu$_{2x}$Y$_{2(1-x)}$O$_3$:Yb scintillating ceramic samples fabricated by Radiation Monitoring Devices Inc. All samples show X-ray excited luminescence peaked at 370 nm. Ultrafast decay time of 1.1 ns was observed by using a microchannel plate-photomultiplier tube-based test bench at Caltech. Considering its intrinsic high density (9.4 g/cm$^3$), Lu$_2$O$_3$:Yb ceramics are promising for future time of fight application for high energy physics experiments.

**Keywords:** scintillators; ceramics; time of flight; ultrafast calorimetry; charge transfer luminescence; fast timing





## 1. Introduction

Inorganic scintillators are widely used in high energy physics (HEP) experiments to construct total absorption electromagnetic calorimeters, providing the best possible energy resolution and position resolution, as well as good electron and photon identification and reconstruction efficiency. The 2019 Department of Energy (DOE) basic research needs (BRN) report [1] points out that ultrafast, radiation hard and cost-effective scintillators are one of the priority research directions for HEP calorimetry. Ultrafast inorganic scintillators are required by future HEP experiments at both the energy and intensity frontiers to mitigate high event rate and pileup. The Compact Muon Solenoid (CMS) experiment is building a barrel timing layer (BTL) consisting of cerium doped lutetium yttrium oxyorthosilicate (Lu$_{2(1-x)}$Y$_{2x}$SiO$_5$:Ce or LYSO) crystals readout by Silicon Photomultipliers (SiPM) for the high luminosity large hadron collider (HL-LHC) [2]. Its timing resolution reaches 30 _ps_. One of the limiting factors for the timing resolution is the 40 ns decay time of LYSO:Ce crystals, which would also cause pile-up in future high-rate experiments. Ultrafast heavy inorganic scintillators with a few nano-second decay time are important to break the pico-second (_ps_) timing barrier for time of flight (TOF) and ultrafast calorimetry applications for future HEP experiments. An example of ultrafast inorganic scintillator is BaF$_2$:Y crystal, which is proposed for the Mu2e-II experiment at Fermilab [2], and also for GHz hard X-ray imaging for future Free-Electron Laser facilities [3–5]. We use two figures of merit for such applications: (1) the light yield in the first ns, and (2) the ratio between the light yield in the first ns and its total light yield (U/T).

Charge transfer (CT) luminescence was observed in Yb$^{3+}$ (4f$^{13}$) activated scintillators [6]. It features two emission bands (CT state —> $^2$F$_{5/2}$) and (CT state —> $^2$F$_{7/2}$) with an energy difference of about 10,000 cm$^{-1}$, and a strong thermal quenching. Depending on the temperature and the composition, ultrafast and fast decay time from sub-nanosecond to tens of nanosecond was observed in Yb$^{3+}$ doped scintillators. Among them, Lu$_2$O$_3$:Yb

shows a high-density (9.42 g/cm$^3$) and large dE/dx (11.6 MeV/cm), so it is attractive for the HEP community. Its high melting point (2490 °C), however, makes the growth of a single crystal expensive. Ceramics are more cost-effective than single crystals because of the following reasons. (1) Ceramic fabrication does not require melting raw material, so can be conducted with a simpler process at a sintering temperature lower than the melting point for single crystals. (2) Ceramic fabrication allows complex shape with minimum after-growth mechanical processing. It thus has a higher raw material usage and low cost. (3) Activator distribution in ceramics is more homogeneous than in crystals by avoiding segregation.

Cost-effective transparent ceramics have been pursued by industry for decades [7]. Previous studies demonstrate that Lu$_2$O$_3$:Yb ceramic plates with optical quality approaching theoretical transmittance can be obtained [8–12]. By fine-tuning the Yb$^{3+}$ doping level, light yield of 500 ph/MeV and decay time of ~1 ns were reported [10,13]. Its light yield in the first ns, however, is still low compared to other candidate ultrafast scintillators. On the other hand, radiation hardness of the scintillators must be investigated for applications in a severe radiation environment, such as the HL-LHC or FCC-hh. Systematic investigation was carried out to understand radiation damage in various inorganic scintillators against γ-rays [14], neutrons [15,16], and protons [16–19]. The radiation hardness of these rare-earth sesquioxide scintillators needs also be checked.

In this investigation, Lu$_2$O$_3$:Yb and Lu$_{2x}$Y$_{2(1-x)}$O$_3$:Yb ((Lu,Y)$_2$O$_3$:Yb) ceramics were fabricated by Radiation Monitoring Devices, Inc, Watertown, MA (RMD). Their optical and scintillation performance were measured at Caltech HEP Crystal Lab. Ultrafast decay time was measured by using a microchannel plate-photomultiplier tube (MCP-PMT)-based test bench. Radiation hardness against γ-rays was also investigated.

## 2. Materials and Methods

Figure 1 shows seven Lu$_2$O$_3$:Yb and Lu$_{2x}$Y$_{2(1-x)}$O$_3$:Yb ((Lu,Y)$_2$O$_3$:Yb) ceramic samples used in this investigation. Table 1 lists their detailed information.

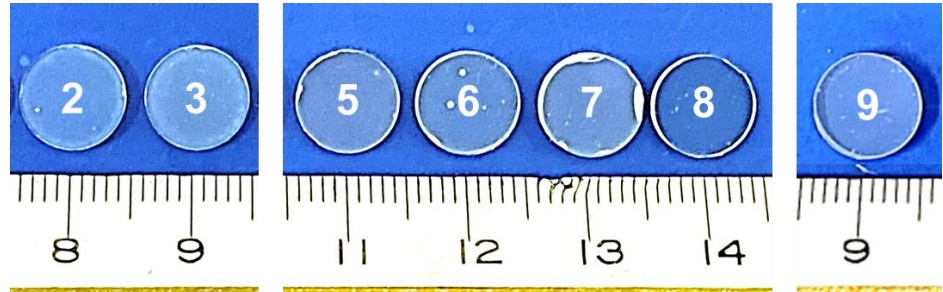

**Figure 1.** A photo showing seven Lu$_2$O$_3$:Yb and (Lu,Y)$_2$O$_3$:Yb ceramic samples fabricated by RMD.

**Table 1.** Dimension and composition of the Lu$_2$O$_3$:Yb and (Lu,Y)$_2$O$_3$:Yb ceramic samples used in this investigation.

| ID | Dimension (mm$^3$) | Composition |
|---|---|---|
| RMD-2 | Φ9 × 1.5 | Lu$_2$O$_3$ |
| RMD-3 | Φ9 × 1 | Lu$_2$O$_3$ |
| RMD-5 | Φ9 × 1.5 | (Lu,Y)$_2$O$_3$ |
| RMD-6 | Φ9 × 1.5 | (Lu,Y)$_2$O$_3$ |
| RMD-7 | Φ9 × 2 | (Lu,Y)$_2$O$_3$ |
| RMD-8 | Φ9 × 1 | Lu$_2$O$_3$ |
| RMD-9 | Φ9 × 2 | (Lu,Y)$_2$O$_3$ |

Their X-ray excited luminescence spectrum (XEL) was measured by using a HITACHI F-4500 spectrophotometer and an Amptek Eclipse-III X-ray tube. Their transmittance

was measured by using a Hitachi U3210 spectrophotometer with 0.2% precision. Their light output (LO) was measured by a Hamamatsu R2059 PMT with a grease coupling for 0.511-MeV $\gamma$-rays from a $^{22}$Na source with a coincidence trigger. The corresponding systematic uncertainty of the light output data is 1%. The sample RMD-2 was irradiated by $\gamma$-rays from a Cs-137 source at Caltech in two steps to reach a total ionization dose (TID) of 10.1 Mrad. This sample was kept at room temperature and wrapped with Al foil to avoid optical bleaching in the entire process. Both transmittance and light output of the sample were measured before and after irradiation at the Caltech HEP crystal lab.

Figure 2 shows the test-benches used to measure the temporal response for (a) BaF$_2$ samples to 511 keV $\gamma$-rays from a $^{22}$Na source with a coincidence trigger, and (b) Lu$_2$O$_3$:Yb ceramic samples to a $^{241}$Am source. A Photek MCP-PMT240 with a rise time and FWHM of 0.18 and 0.82 ns, respectively, was used to measure the scintillation signal. A 2.5 GHz Agilent MSO 9254A was used to collect and process the signal.

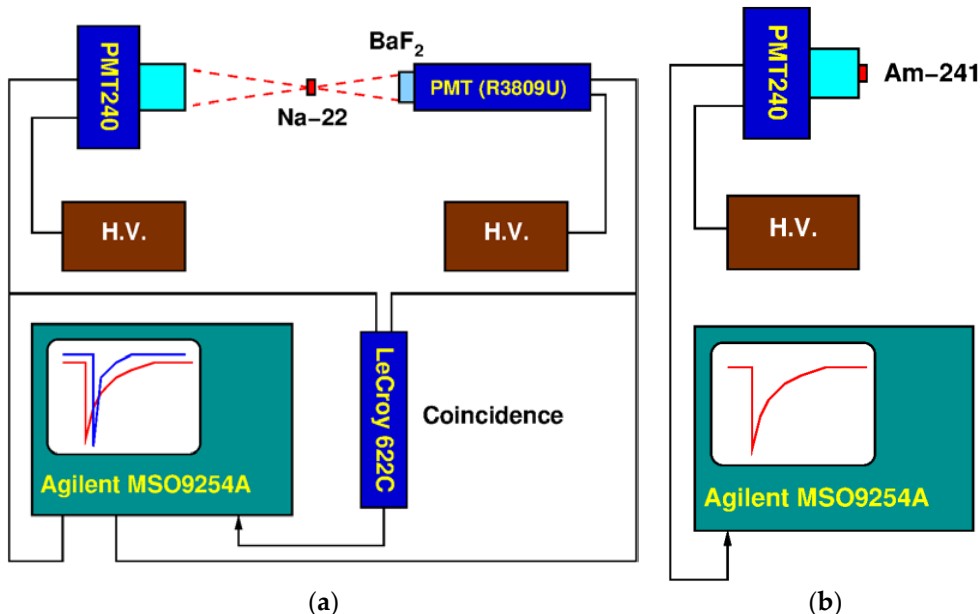

**Figure 2.** A schematic showing MCP-based test benches for temporal response measurement for (**a**) BaF$_2$ samples to 511 keV $\gamma$-rays from a $^{22}$Na source with a coincidence trigger, and (**b**) Lu$_2$O$_3$:Yb ceramic samples to a $^{241}$Am source.

Rise time and decay time of the measured scintillation pulse was obtained by fitting the temporal response of the pulse shape with the following equation [20]:

$$V\left(t\right) = A\left(e^{-\frac{t}{\tau_d}} - e^{-\frac{t}{\tau_r}}\right) + B,\tag{1}$$

where $V$ is the measured pulse amplitude, $B$ represents background noise, and $\tau_r$ and $\tau_d$ are respectively the rise time and decay time. The full width at half maximum (FWHM) of the pulse was calculated by the fitting.

The data presented in this paper are not corrected by the Instrument Response Function.

## 3. Results and Discussions

Figure 3 shows X-ray excited luminescence spectra measured for the Lu$_2$O$_3$:Yb ceramic sample 2 (top) and 3 (middle), and the (Lu,Y)$_2$O$_3$:Yb ceramic sample 6 (bottom). All three samples show consistent XEL peaked at ~370 nm. A slight difference was observed for sample 6, which can be attributed to the Y admixture.

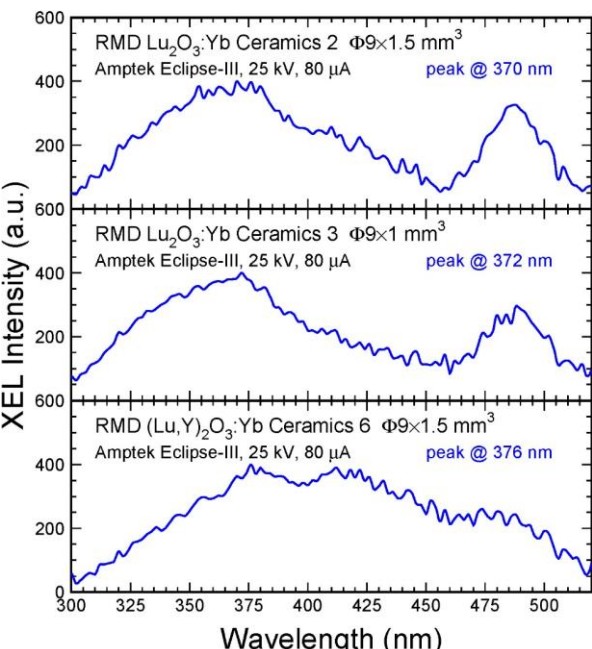

**Figure 3.** XEL spectra measured for the $Lu_2O_3$:Yb ceramic sample 2 (**top**) and 3 (**middle**), and the $(Lu,Y)_2O_3$:Yb ceramic sample 6 (**bottom**).

Figure 4 shows transmittance spectra measured for the $Lu_2O_3$:Yb ceramic samples 2 (top) and 3 (middle), and the $(Lu,Y)_2O_3$:Yb ceramic sample 6 (bottom). The XEL spectra (blue dash lines) are also shown in the figure, as well as the theoretical limit of transmittance (black dots) and the numerical values of the emission weighted longitudinal transmittance (EWLT). The theoretical limit of transmittance is calculated by using the refractive index assuming multiple bounces and no internal absorption. The EWLT value represents the numerical value of transmittance over emission spectrum. The $(Lu,Y)_2O_3$:Yb sample 6 shows poor transmittance. This is due to the scattering centers in the sample.

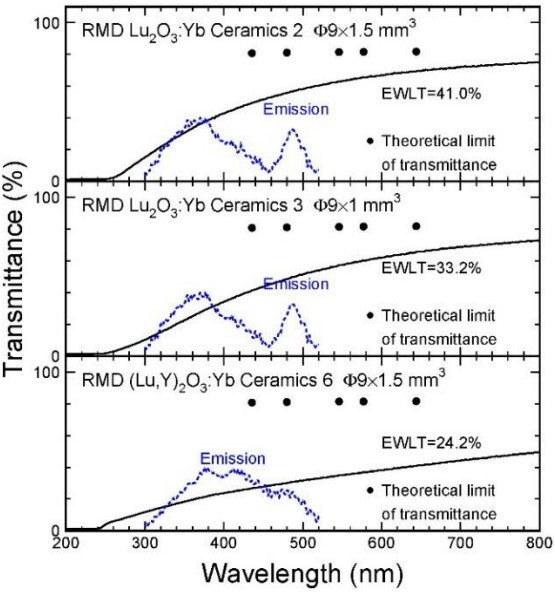

**Figure 4.** Transmittance spectra (black lines) measured for the $Lu_2O_3$:Yb ceramic samples 2 (**top**) and 3 (**middle**), and the $(Lu,Y)_2O_3$:Yb ceramic sample 6 (**bottom**).

Figure 5 shows LO as a function of integrated time measured for the $Lu_2O_3$:Yb ceramic samples 2 (top) and 3 (middle), and the $(Lu,Y)_2O_3$:Yb sample 9 (bottom). Taking out the emission-weighted quantum efficiency (EWQE) of 20%, these $Lu_2O_3$:Yb samples shows light yield of up to 280 ph/MeV with negligible slow component. By using the LO data and the decay time from Figure 6 below the corresponding light yield in the first ns and the U/T ratio are 170 photons/MeV and 61% respectively for the $Lu_2O_3$:Yb ceramic sample 3, which is very promising. On the other hand, doping with yttrium is found to increase the light output, but also introduce slow light with decay time of ~100 and ~2500 ns.

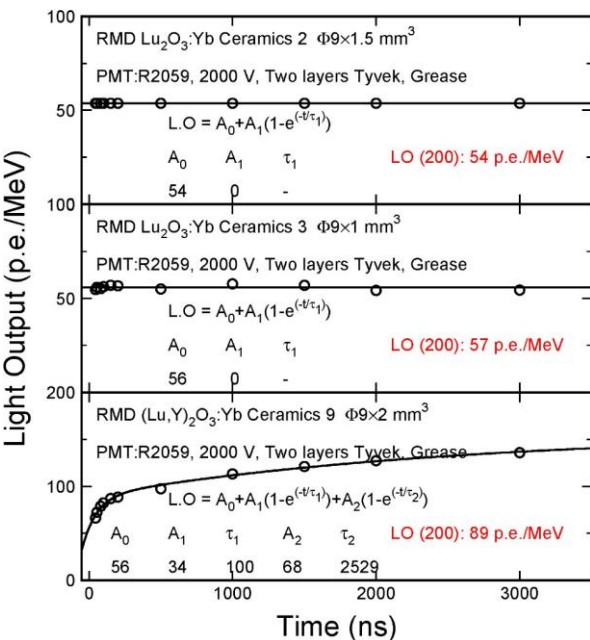

**Figure 5.** Light output is shown as a function of integration time for the $Lu_2O_3$:Yb ceramic samples 2 (**top**) and 3 (**middle**), and the $(Lu,Y)_2O_3$:Yb ceramic sample 9 (**bottom**).

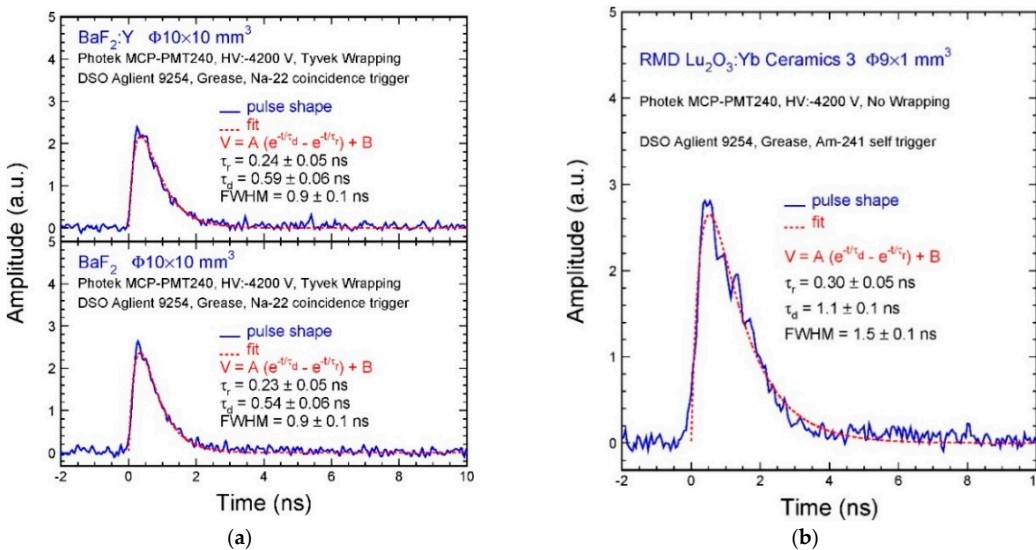

**Figure 6.** Temporal response measured by a Photek MCP-PMT240 for (**a**) BaF2 samples and (**b**) the $Lu_2O_3$:Yb ceramic sample 3.

Figure 6 shows the temporal response measured by a Photek MCP-PMT240 for (a) a $BaF_2$:Y crystal (top) and a $BaF_2$ crystal (bottom) and (b) one $Lu_2O_3$:Yb ceramic sample with an ultrafast decay time of 0.6 and 1.1 ns, respectively. The corresponding rise time and

FWHM are 0.2 and 0.9 ns for BaF$_2$ crystals, and 0.3 and 1.5 ns for Lu$_2$O$_3$:Yb. These values can be compared to the MCP-PMT240 response of 0.18 and 0.82 ns, respectively.

Figure 7 shows (a) transmittance and (b) light output as a function of the integral time measured before and after $\gamma$-ray irradiation with a total dose of 5.1 Mrad (red) and 10.1 Mrad (blue) for the Lu$_2$O$_3$:Yb ceramic sample 2. Radiation damage appears approaching saturation after 5.1 Mrad. Light output and transmittance loss can be attributed to the radiation induced absorption in the Lu$_2$O$_3$:Yb sample. Damage recovery and dose rate dependence will be studied for thicker Lu$_2$O$_3$:Yb ceramics with better optical quality to reduce the uncertainty in the radiation induced absorption data to facilitate direct comparison with other well-investigated crystal scintillators [14].

Table 2 compares the scintillation performance for various fast and ultrafast inorganic scintillators [4]. Among them, Lu$_2$O$_3$:Yb ceramics show the highest density and dE/dx, and the shortest radiation and nuclear interaction length. Its ultrafast decay time of 1.1 ns and the high U/T ratio of 61% make it promising for future TOF and ultrafast calorimetry applications. Additional work is needed to improve its light yield in the first ns while keeping the slow component under control.

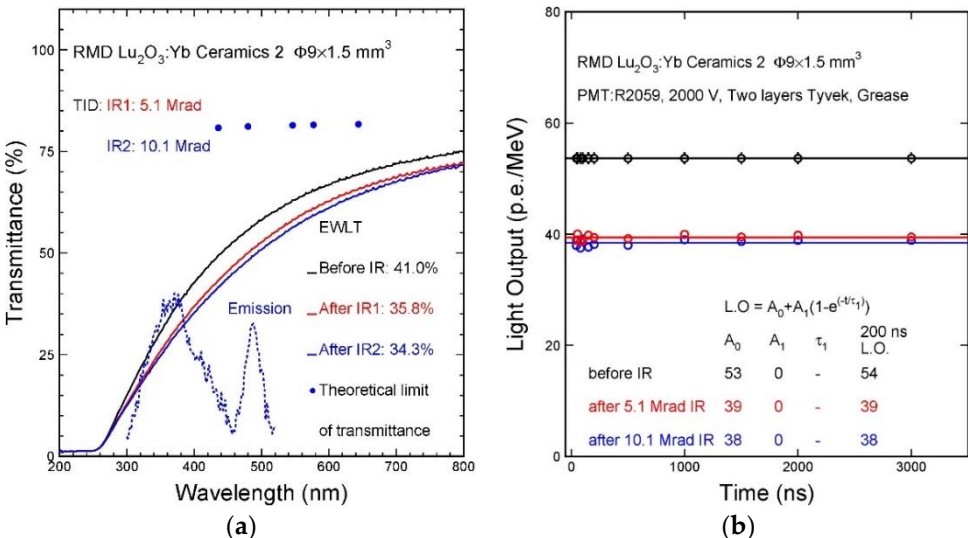

**Figure 7.** (**a**) Transmittance and (**b**) light output as a function of integral time measured before and after irradiation are shown for the Lu$_2$O$_3$:Yb ceramic sample 2 before and after the total $\gamma$-ray dose of 5.1 (red) and 10.1 Mrad (blue).

**Table 2.** Scintillation performance of various fast and ultrafast inorganic scintillators.

| | BaF$_2$ | BaF$_2$:Y | ZnO:Ga | Lu$_2$O$_3$:Yb | YAP:Yb | YAG:Yb | β-Ga$_2$O$_3$ | PWO | LYSO:Ce | LuAG:Ce | YAP:Ce | GAGG:Ce | LuYAP:Ce | YSO:Ce |
|---|---|---|---|---|---|---|---|---|---|---|---|---|---|---|
| Density (g/cm$^3$) | 4.89 | 4.89 | 5.67 | 9.42 | 5.35 | 4.56 | 5.94 | 8.28 | 7.4 | 6.76 | 5.35 | 6.5 | 7.2 [6] | 4.44 |
| Melting points (°C) | 1280 | 1280 | 1975 | 2490 | 1870 | 1940 | 1725 | 1123 | 2050 | 2060 | 1870 | 1850 | 1930 | 2070 |
| X$_0$ (cm) | 2.03 | 2.03 | 2.51 | 0.81 | 2.59 | 3.53 | 2.51 | 0.89 | 1.14 | 1.45 | 2.59 | 1.63 | 1.37 | 3.10 |
| R$_M$ (cm) | 3.1 | 3.1 | 2.28 | 1.72 | 2.45 | 2.76 | 2.20 | 2.00 | 2.07 | 2.15 | 2.45 | 2.20 | 2.01 | 2.93 |
| λ$_I$ (cm) | 30.7 | 30.7 | 22.2 | 18.1 | 23.1 | 25.2 | 20.9 | 20.7 | 20.9 | 20.6 | 23.1 | 21.5 | 19.5 | 27.8 |
| Z$_{eff}$ | 51.0 | 51.0 | 27.7 | 67.3 | 32.8 | 29.3 | 27.8 | 73.6 | 63.7 | 58.7 | 32.8 | 50.6 | 57.1 | 32.8 |
| dE/dX (MeV/cm) | 6.52 | 6.52 | 8.34 | 11.6 | 7.91 | 7.01 | 8.82 | 10.1 | 9.55 | 9.22 | 7.91 | 8.96 | 9.82 | 6.57 |
| λ$_{peak}$ [1] (nm) | 300<br>220 | 300<br>220 | 380 | 370 | 350 | 350 | 380 | 425<br>420 | 420 | 520 | 370 | 540 | 385 | 420 |
| Refractive Index [2] | 1.50 | 1.50 | 2.1 | 2.0 | 1.96 | 1.87 | 1.97 | 2.20 | 1.82 | 1.84 | 1.96 | 1.92 | 1.94 | 1.78 |
| Normalized Light Yield [1,3] | 42<br>4.8 | 1.7<br>4.8 | 6.6 [4] | 0.95 | 0.19 [4] | 0.36 [4] | 6.5<br>0.5 | 1.6<br>0.4 | 100 | 35 [5]<br>48 [5] | 9<br>32 | 190 | 16<br>15 | 80 |
| Total Light yield (ph/MeV) | 13,000 | 2000 | 2000 [4] | 280 | 57 [4] | 110 [4] | 2100 | 130 | 30,000 | 25,000 [5] | 12,000 | 58,000 | 10,000 | 24,000 |
| Decay time [1] (ns) | 600<br>0.5 | 600<br>0.5 | <1 | 1.1 [4] | 1.5 | 4 | 148<br>6 | 30<br>10 | 40 | 820<br>50 | 191<br>25 | 570<br>130 | 1485<br>36 | 75 |
| LY in 1st ns (photons/MeV) | 1200 | 1200 | 610 [4] | 170 | 28 [4] | 24 [4] | 43 | 5.3 | 740 | 240 | 391 | 400 | 125 | 318 |
| LY in 1st ns /Total LY (%) | 9.2 | 60 | 31 | 61 | 49 | 22 | 2.0 | 4.3 | 2.5 | 1.0 | 3.3 | 0.7 | 1.3 | 1.3 |
| 40 keV Att. Leng. (1/e, mm) | 0.106 | 0.106 | 0.407 | 0.127 | 0.314 | 0.439 | 0.394 | 0.111 | 0.185 | 0.251 | 0.314 | 0.319 | 0.214 | 0.334 |

[1] top/bottom row: slow/fast component. [2] at the emission peak. [3] normalized to LYSO:Ce. [4] excited by alpha particles. [5] ceramic with 0.3 Mg at% co-doping. [6] density for composition Lu$_{0.7}$Y$_{0.3}$AlO$_3$:Ce.

## 4. Conclusions

Development of ultrafast heavy crystals with sub-nanosecond decay time is important to break the *ps* timing barrier for future HEP TOF system and ultrafast calorimetry, and for GHz hard X-ray imaging. All $Lu_2O_3$:Yb and $(Lu,Y)_2O_3$:Yb samples from RMD show XEL emission peaked at ~370 nm. $Lu_2O_3$:Yb ceramics show light yield up to 280 ph/MeV with negligible slow component. Mixing $Lu_2O_3$ with $Y_2O_3$ increases light yield but introduces a significant slow component of 100 and 2500 ns decay time. A sub-nanosecond decay time of 1.1 ns was measured by using MCP-PMT. With a high density, an ultrafast decay time, a light yield in the first nanosecond of 170 photon/MeV and an U/T ratio of 61%, $Lu_2O_3$:Yb ceramics are promising for future TOF and ultrafast calorimetry applications. This investigation will continue to optimize the composition of $Lu_2O_3$:Yb and increase its transparency and ultrafast light while keeping the slow component under control.

**Author Contributions:** Conceptualization, R.-Y.Z. and L.S.P.; methodology, R.-Y.Z., L.Z. and L.S.P.; investigation, C.H., L.Z., R.-Y.Z., L.S.P, Y.W. and J.G.; resources, R.-Y.Z., L.S.P., Y.W. and J.G.; writing—original draft preparation, C.H.; writing—review and editing, R.-Y.Z., L.Z., L.S.P. and J.G. All authors have read and agreed to the published version of the manuscript.

**Funding:** This research was funded by the Department of Energy, Office of Science, Office of High Energy Physics, under Award Number DE-SC0011925 and the Small Business Innovation Research program under Award Number DE-SC0021686.

**Data Availability Statement:** All data are available on the Web. The data presented in this paper are openly available in: http://www.hep.caltech.edu/~zhu/ accessed on 9 August 2022.

**Conflicts of Interest:** The authors declare no conflict of interest.

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
