# Peer review of "Novel Ultrafast Lu2O3:Yb Ceramics for Future HEP Applications"

_instruments, doi:10.3390/instruments6040067_

Round 1
Reviewer 1 Report
The article presents the fundamental characterization of Ytterbium-doped Lu2O3 and (Lu,Y)2O3 ceramics scintillators, produced by RMD Inc, and performed at Caltech HEP Crystal Laboratory. In particular, their output spectrum, transmittance, and timing properties were measured; additionally the effects of ionizing radiation (up to ~10 Mrad) on these parameters were assessed.
The article is extremely well written and clear, it presents relevant results. The authors showcase an acute grasp on the subject. The performed measurements represent fundamental steps in the characterization of such devices. The effort is justified by the demand of heavy and ultrafast inorganic scintillators, in particular in the framework of HEP experiments (such as CERN-CMS). As stated by the authors, these type of scintillators require additional investigation before being considered for such appilcations, although this article can serve as a starting point, as well as reference for other efforts. The experimental procedures followed for this characterizations are well described, establised, and adequate for the case.
I would suggest some minor adjustment:
- line 97: I think replacing V with V(t) in the equation would make it more clear;
- line 104: I suggest to slightly expand the comment on Fig.3 , maybe with a remark abut the reasons for the different behaviour between samples 2-3 and 6, in terms of their spectra;
- line 119: referencing the fitting equation would make it easier for the reader to follow;
- line 125: regarding the slower component present in the Y-doped sample, do you expect this to be a deal-breaker for its employment in HEP?
- line 129: Are the measured timing constants visibly affected by the sample thickness or not at all?
- line 135 (Fig. 6): I suggest showing the two plots with the same range in the x-axis for a more equal comparison; how was the fitting range chosen for the two signals? Do you have an explanation for the visible ripples in the signals?
- A comment on the expected energy resolution (if available) compared to a reference scintillator.
Reviewer 2 Report
Dear Authors,
Regarding the manuscript “Novel Ultrafast Lu2O3:Yb Ceramics for Future HEP Applications”. In my opinion the performed studies are very interesting and valuable for the community. Also the technical merit is good and the results seem convincing.
Generally, I slightly miss more thorough descriptions and discussions of the measurements and setups. For example, it is not completely clear if the measured transmission relates to the 1.5 mm thickness? The temporal response was measured with an MCP-PMT, which imposes certain limits on the precision of the values, i.e. the impulse response function can be quite large in this setup and should be mentioned/discussed.
Regarding Table 2 in my opinion a lot of references for the given values are missing, or did the authors measure all these crystals and values by themselves? Some values in this table also seem to be outdated, as newer publications show different magnitudes (although I have to admit it is in the ballpark).
Overall I find the presented scintillation materials and its studies very interesting and I think the paper is suitable for publication.
Please find some more small comments below:
Page/Line number: Comment
1/11: Abstract: Mentioning the advantages of ceramics I immediately think about their optical properties. Maybe a short statement could be given after the sentence: “Inorganic scintillators in ceramic ….”
1/13 ff: Generally, I would write “X-ray” with capital letter in the beginning. It is up to the authors, but now there is both ways in the manuscript.
1/34: If you are talking about pile-up effects the 40ns decay time of LYSO:Ce might be indeed a limiting factor. For the timing resolution this is not necessarily true, as the light yield of LYSO:Ce is very high (40 000 ph/MeV). The figure of merit for timing is decay time over light yield. Hence, for LYSO:Ce one obtains 40ns/40 000ph/MeV=1/1000 which is quite fast.
2/58: Related to my comment and figure of merit above the time resolution obtainable with Lu2O3:Yb is two times worse. But indeed it is very fast and if there are no slower components present also quite interesting for HEP applications.
Equation 1: Does a reference to the equation make sense or do the authors see it as common knowledge?
Figure 4: How was the theoretical limit of the transmittance calculated (simple Fresnel equations)? Maybe you can state this in the text. I was also wondering what the transmittance on the y-scale represents? Is it the measured transmittance with 1.5mm thickness? If this is the case, I was wondering whether the optical properties of the samples are relatively poor?
5/121: I think in the sentence is a “relatively” after the 82% missing. Besides that, I have a slight problem with the U/T ratio, as the 1 ns for U seems to be chosen a bit arbitrary? In my opinion the true figure of merit for timing is the light yield over the decay time (assuming the rise time is almost zero). If the rise time is not zero it should multiply with the decay time. The U/T ratio is something similar, but in my view more complex than just focusing on the light yield, rise time and decay time.
5/133 and figure 6: What is the impulse response function (IRF) of the setup? It is probably mainly defined by the single photoelectron shape of the MCP-PMT (and also time resolution of the reference detector). Hence, the rise times stated are most likely only determined by the system IRF and not the crystal, as I do not see any signs that the IRF was taken into account in the fit. If this is true, the rise time values are wrong (overestimated) and most likely the decay time values too. There are also reflections (cable not properly terminated?) seen in the plots. In the end I think these plots are only a guidance to show that the material is fast. I think this should be discussed in the text.
Figure 7: What is EWLT? Did I miss the definition in the text? Is the light output in subfigure b only determined by the lower transmittance or is the scintillation efficiency also altered by irradiation? It could be good to discuss this in the text.
Table 2: As already mentioned, in my opinion some references are missing.
Kind regards
